# Intake of Dietary Salicylates from Herbs and Spices among Adult Polish Omnivores and Vegans

**DOI:** 10.3390/nu12092727

**Published:** 2020-09-06

**Authors:** Danuta Gajewska, Paulina Katarzyna Kęszycka, Martyna Sandzewicz, Paweł Kozłowski, Joanna Myszkowska-Ryciak

**Affiliations:** Department of Dietetics, Institute of Human Nutrition Sciences, Warsaw University of Life Sciences (WULS), 159C Nowoursynowska Str, 02-776 Warsaw, Poland; paulina_keszycka@sggw.edu.pl (P.K.K.); martyna.sandzewicz@gmail.com (M.S.); kozlowski.pawel96@gmail.com (P.K.); joanna_myszkowska_ryciak@sggw.edu.pl (J.M.-R.)

**Keywords:** herbs, spices, dietary salicylates, intake, omnivores, vegans

## Abstract

There is a growing body of evidence that a diet rich in bioactive compounds from herbs and spices has the ability to reduce the risk of chronic diseases. The consumption of herbs and spices is often overlooked in the studies on food intake. However, measurement of dietary intake of these products, as a source of bioactive compounds, including salicylates, has recently gained much significance. The aims of the study were (i) to assess the intake of herbs and spices at the individual level and (ii) to calculate the dietary salicylates intake from herbs and spices among adult omnivores and vegans. The study group consisted of 270 adults aged 19 to 67 years, including 205 women and 65 men. Among all, 208 individuals were following an omnivorous diet while 62 were vegans. A semi-quantitative food frequency questionnaire (FFQ) was designed to assess the habitual intake of 61 fresh and dried herbs and spices during the preceding three months. The five most frequently eaten herbs among omnivores were parsley, garlic, dill, marjoram and basil, while among vegans they were garlic, parsley, ginger, basil and dill. An average intake of all condiments included in the study was 22.4 ± 18.4 g/day and 25.8 ± 25.9 g/day for both omnivores and vegans, respectively (*p* = 0.007). Estimated medial salicylates intake was significantly higher among vegans (*p* = 0.000) and reached 5.82 mg/week vs. 3.13 mg/week for omnivores. Our study confirmed that herbs and spices are important sources of salicylates; however, the type of diet influenced their level in the diet. Vegans consume significantly more total salicylates than omnivores.

## 1. Introduction

Herbs and spices are commonly used in cooking to increase the flavor and the taste of dishes. The types of herbs and spices consumed vary considerably across cultures and countries. The highest consumption of these products is observed in Asian countries (such as India, Thailand and China), South Africa and Latin America [1,2]. In contrast, the western-style diet is characterized by a high intake of salt and sugar, and it is considered rather spice-free. However, the popularity of oriental cuisine and a growing preference for ethnic or spicy food raised the consumption of spices in Europe and North America [2,3]. Data indicate that the EU market is the largest market for herbs and spices in the world and the leading consumers (among EU citizens) are Germans, British, Romanians and Hungarians [4]. Among the factors influencing the popularity of herbs, spices and spice blends (especially organic) are that they are promoted as natural, “zero calories superfood” by bloggers and celebrity chefs [5,6].

National cuisines and ethnic cuisines differ in terms of typical herbs added to dishes. Spices typical for Indian cuisine are turmeric, saffron, cardamom, cinnamon, cloves, garlic, peppers, cumin and coriander. Mediterranean-style cuisine is famous for parsley, oregano, basil, thyme, dill, fennel, marjoram, rosemary, bay leaf, lavender, sage, savory, cumin and coriander [3,7]. In China, the most popular condiments include peppers, peppercorns, ginger, fennel, cloves, star anise and cinnamon, whereas Italian and Greek cuisine is rich in basil, garlic and oregano. The typical spices of Arabic cuisine are saffron, peppers, allspice, turmeric, garlic, cumin, cinnamon, parsley and coriander, while “the queen” of Hungarian cuisine is paprika powder [3]. Whereas, Polish cuisine is characterized by a mix of strong flavors of strong flavors: pungent (horseradish, mustard, chives, onion, garlic), and sour (pickled cucumbers, cabbage), spices (pepper, juniper, caraway), herbal (dill, parsley) and sweet (honey) [8].

Unfortunately, the consumption of herbs and spices in Europe (despite a growing trend) seems to be small and is often overlooked in the studies on food intake. However, measurement of the dietary intake of these products, as a source of bioactive compounds, including salicylates, has recently gained much significance [3,9,10]. Today, the importance of herbs and spices is being acknowledged not only because of their culinary properties, but mainly because of their potential health benefits. There is a growing body of evidence that a diet rich in bioactive compounds from herbs and spices has the ability to reduce the risk of chronic diseases, including cancer, inflammatory diseases, cardiometabolic diseases, atherosclerosis and diabetes [11,12,13,14]. The use of herbs and spices in regular meals may have beneficial effects not only on cardio-vascular function, the initiation and development of atherosclerosis and reduction of the risk of CVD (Cardiovascular Disease), but also on the modulation of gastro-intestinal function, the reduction of chronic inflammation, and even on slimming and optimization of skeletal muscle performance [3,15,16].

Several studies demonstrate therapeutic effects of phytochemicals from herbs and spices. These effects include, among others: hypotensive effects of turmeric, star anise, saffron, chili pepper, garlic and cinnamon [3,17]; anti-inflammatory effects of turmeric, ginger and cinnamon [11,18]; antibacterial and antifungal effects of cinnamon, clove, cumin, garlic, basil, oregano, thyme, turmeric and ginger [19,20,21]; antioxidant activities of clove, chili, cinnamon, cumin, ginger, oregano, sage, turmeric, marjoram, nutmeg and thyme [2,13,21,22,23]; antidiabetic properties of bay leaf, cinnamon, cumin, wormwood, fenugreek, mustard, marjoram and thyme [24]; immunomodulatory action of turmeric [25]; anti-obesity effects (including body weight and lipid-lowering effects) of carnosic acid in rosemary and sage and capsaicinoids in red chili pepper [26]; lipid-lowering effect of basil, fenugreek, curcumin, ginger, onion and garlic [13] and anti-cancer properties of cloves, black cumin, black pepper, basil, chili pepper, garlic, ginger, onion, oregano, saffron, turmeric, thyme and marjoram [27,28].

Despite the large scale of literature describing the beneficial health effects of herbs and spices, the research describing the intake of salicylates from these food products are limited [29,30,31]. Herbs and spices were found to be the second major dietary sources of salicylates in the Scottish population (17%) [30] but in the diet of Poles they only contributed 7% of the total amount of salicylates [31]. Several studies confirmed that salicylic acid (SA) (C_7_H_6_O_3_, 2-hydroxybenzoic acid) and its derivatives (salts and esters) may have pro-health properties [29,30,32]. The beneficial effects of SA and dietary salicylates comprise anti-inflammatory, antiatherogenic, anticancer and antithrombotic properties, as well as cardioprotective effects. However, the therapeutic dose of SA from dietary sources, which could be used for disease prevention, has not yet been determined [33,34]. Unfortunately, in some sensitive patients, even a small dose of dietary salicylates may cause serious health problems including asthma, nasal polyps, rhinitis, urticarial, angioedema or gastrointestinal symptoms [29,30,32]. SA is one of the most common pseudo allergens, widely distributed in plant-based foods, including herbs and spices. Therefore, information on salicylates content in food is important for dietitians as well as their patients, as it helps to formulate adequate recommendations and to plan a diet requiring dietary salicylate avoidance.

The hypothesis of the current study was that, regardless of the dietary pattern, herbs and spices are important sources of salicylates. The aims of the study were (i) to assess the intake of herbs and spices at the individual level and (ii) to calculate the dietary salicylates intake from herbs and spices among adult omnivores and vegans. To estimate the intake of salicylates we used several database of salicylates contents in food [30,32,33,34,35,36,37] including our own studies [9,38,39].

## 2. Materials and Methods

### 2.1. General Information

Data collection was carried out from September 2018 through September 2019. All procedures involving human subjects were conducted according to the guidelines laid down in the Declaration of Helsinki and approved by the Ethics Committee of the Faculty of Human Nutrition of the Warsaw University of Life Sciences—SGGW (Resolution No. 33/2018). Inclusion criteria were an adult age and consent to complete the study questionnaire.

### 2.2. Participants and Study Design

The study was the cross-sectional, prospective, non-randomized trial aims to compare the intake of dietary salicylates from herbs and spices among two different subgroups—omnivores and vegans. The study subjects were recruited using snowball sampling among students of the Faculty of Human Nutrition of the Warsaw University of Life Sciences (Poland) and their family and friends. Additionally, vegan individuals were recruited using virtual snowball sampling among vegan social networking sites. The study group consisted of 270 adults aged 19 to 67 years, including 205 women and 65 men. Among all, 208 individuals were following an omnivorous diet while 62 were vegans. The vegan subjects were on a vegan diet for a period of 0.5 to 10 years.

We have used both the Computer-Assisted Web Interviewing (CAWI) and the Pen-and-Paper Personal Interview (PAPI) methods to collect the information. After enrollment, all participants were given the questionnaire form and clear written or oral instructions in how to complete the recording of herbs and spices consumption. Questionnaires were provided by a trained interviewer. Voluntary completion of the questionnaire was considered to imply informed consent according to the Helsinki Declaration guidelines. The data received by PAPI method have been entered into an Excel database (Microsoft Office Professional Plus 2016, Katowice, Poland) manually.

### 2.3. Socio-Demographic and Anthropometric Data

The questionnaire collected information on sociodemographic and anthropometric variables, physical activity, health status and perception of diet quality. Self-reported body weight and height of individuals were used to calculate body weight status. Body Mass Index (BMI) was calculated as weight in kg divided by height in square meters. According to the WHO standard, BMI criteria were applied for individuals as follows: BMI < 18.5 kg/m^2^—underweight; BMI 18.5–24.9 kg/m^2^—normal body weight; BMI 25–29.9 kg/m^2^—overweight and BMI ≥ 30 kg/m^2^—obesity [40].

### 2.4. The Herbs and Spices Intake

The list of herbs and spices included in our study was established based on several factors, including: the assortments of these products in four major supermarkets in Warsaw (Poland), the sales information from three major manufacturers of herbs and spices in Poland, as well as the analysis of customers’ food preferences from our previous study [31,38].

The structured list contained within 61 individual types of herbs and spices, categorized into three groups: 1. sixteen fresh herbs: basil, chili, coriander (cilantro), dill, garlic, ginger, lovage, marjoram, melissa, oregano, parsley, peppermint, rosemary, sage, tarragon, thyme; 2. thirty one dried spices: anise, allspice, basil, bay leaf, black cumin, black pepper, caraway, cardamom, cayenne pepper, celery, cloves, cocoa, cumin, fenugreek, flax seeds, garlic, ginger, hot paprika, juniper, lovage, mustard, nutmeg, onion, oregano, rosemary, saffron, savory, sweet paprika, turmeric, white pepper; 3. fourteen others seasonings or condiments: curry spice blend, horseradish, herbs de Provence (thyme, rosemary, bay, basil, savory), Maggi, mustard, ready-mix spices to dishes, soy sauce, Vegeta, salt, lemon juice, citric acid, vinegar, balsamic vinegar and wine vinegar.

Herb and spice intake was assessed based on the frequency of the intake and portion size per eating occasion. We have used the FAO (Food and Agricultural Organization) step-by-step guide to select an appropriate diary assessment method [41].

A validated semi-quantitative food frequency questionnaire (FFQ) was designed to assess the habitual intake of fresh and dried herbs and spices during the preceding 3 months among adults. The responders were asked to report on the frequency of herbs and spices (listed above) consumed both separately, and as integrated parts of prepared dishes.

Frequencies of consumption were measured in ten categories. The options of frequency of consumption were as follows: “1”, “2” or “≥3 times per day”; “1–2 times per week”, “3–4 times per week”, “≥5 per week”; “less than once a month”, “1–3 times per month”; “rarely” and “never”. Each frequency category was assigned a value equal to the midpoint of the range specified in that category. The options for portion sizes were described in household units as “teaspoon”, a “half of teaspoon”, a “large pinch” and a “small pinch”. A “small pinch” was defined as the amount of herbs and spices that can be taken with two fingers (thumb and index finger), while a “large pinch” was define as the amount that can be taken with 3 fingers (thumb, index and middle). A pinch is a very small amount (much less than a teaspoon) and not formally defined. In order to compile the database of portion size we have weighed on a scales (Radwag WAA 100/C/1, with 0.001 g readability) small and large pinches, as well as a teaspoon and a half of teaspoon of all herbs and spices included in our study. All measurements were done in three replications and the means were used for the calculation. After collecting data on the frequency of the consumption of herbs and spices, the results previously expressed in household units, were converted into weekly consumption in grams. If the respondent chose the frequency of consumption at the “rarely or never” level, the value of weekly consumption was considered to be 0.

### 2.5. The Salicylates Intake

To calculate total salicylates intake (free salicylic acid plus bound salicylic acid), we multiplied the daily intake of herbs and spices (g/day) by the salicylates content of 33 condiments item. We have used several databases on salicylates content in food, including the recent own study [9,38,39] as well as other studies [30,32,33,34,35,36,37].

### 2.6. Statistical Analysis

All statistical analyses were conducted using Statistica version 13.1 (Copyright©StatSoft, Inc, 1984–2014, Cracow, Poland). FFQ was validated prior to research. The reproducibility of FFQ has been assessed by administering it twice to the omnivore group (*n* = 20, Spearman rank correlation coefficients varied between 0.49 and 0.71). The agreement between food frequency responses with 3-day food records as a reference method was assessed among vegans (*n* = 20) using the Bland–Altman method (the Bland–Altman index for examined condiments ≤ 5%).

Categorical variables were presented as a sample percentage (%), while continuous variables were expressed as the mean, standard deviation and quartiles in the total group and according to dietary patterns (omnivores vs. vegans). The intake of individual herbs and spices was presented as a frequency of intake in times per day/week/month and as amount in grams per week. The normality of the continuous variables’ distribution was assessed with the Shapiro–Wilk test. All data concerning intake of herbs and spices showed skewed distribution. The median values for the majority of herbs and spices were zero. These values did not provide much useful information about the amount of individual herb and spice intake. Therefore, we decided to present data as an average value as well as a median, knowing that the median values are more appropriate estimates for skewed data. Statistical significances for nominal variables were determined with the Pearson’s chi-square test, for non-normally distributed variables the Mann–Whitney U test was used. The correspondence analysis was used to study the relationship between the type of diet and selected parameters (gender, body weight status, physical activity level, occupation, self-assessment of diet regularity and health status). For all tests, *p* < 0.05 was considered as significant.

## 3. Results

Our study group consisted of 270 adults with a mean age 37.0 and 28.11 years old among omnivores and vegans, respectively. Table 1 presents the characteristics of subjects from two study groups. All demographic variables differ significantly between omnivores and vegans. Vegans tended to be younger, more physically active and had a lower body mass. On the other hand, subjects in omnivore group were more likely to be older, less physically active and had higher body mass. Moreover, vegans esteemed their health and diet better (Figure 1). The BMI of omnivores was significantly higher compared with the BMI of vegans; however, in both groups the majority were individuals with normal body weight.

### 3.1. Frequency and Amount of Intake of Fresh Herbs and Spices

Table 2 presents the frequency of consumption of fresh herbs, whereas Table 3 presents the intake of these products among both study groups. The analysis of consumption frequency found that parsley, garlic, dill, marjoram and basil were most commonly consumed herbs among omnivores, while garlic, parsley, ginger, basil and dill among vegans. All vegans (100%) and 83.6% of the omnivores consumed fresh garlic at least once a month. In turn, the least consumed herbs were tarragon, lovage and sage among vegans and sage, tarragon and coriander among omnivores.

Median consumption of fresh herbs ranged from 0 to 23.48 g/week (garlic) among omnivores, while among vegans these values rage from 0 to 49.31 g/week (garlic). We found significant differences in the intake of some herbs between groups, including fresh basil (*p* = 0.000), coriander (*p* = 0.000), garlic (*p* = 0.002), ginger (*p* = 0.000) and marjoram (*p* = 0.000).

### 3.2. Frequency and Amount of Intake of Dried Herbs and Spices

As shown in Table 4, black pepper, sweet paprika, bay leaf, allspice and hot paprika were found to have the highest consumption frequency among omnivores. Black pepper, turmeric, oregano, sweet paprika and basil were most commonly consumed among vegan respondents. More than 80% of the participants from both groups consumed black pepper at least once a week, and more than 30% at least once a day. Saffron, one of the world’s most costly spices, was the less popular spice among both omnivores and vegans. Median intakes were observed to be above 0 g only for allspice, basil, bay leaf, black pepper, cocoa, hot paprika, oregano and sweet paprika among omnivores. In the vegan group these were allspice, basil, bay leaf, black pepper, Cayenne pepper, cloves, cocoa, cumin, flax seed, oregano, rosemary, sweet paprika and turmeric (Table 5). From all 31 dried herbs and spices, the intake of 12 of them differed significantly between the study groups. It concerned allspice (*p* = 0.000), basil (0.004), bay leaf (0.000), caraway (*p* = 0.008), cardamom (*p* = 0.006) Cayenne pepper (*p* = 0.000), cloves (*p* = 0.000), cumin (*p* = 0.000), flax seed (*p* = 0.000), oregano (*p* = 0.020), rosemary (*p* = 0.000) and turmeric (*p* = 0.000).

### 3.3. Frequency and Amount of Intake of Other Seasoning

The frequency of consumption of seasoning blends and other condiments is presented in Table 6. Table 7 presents the intake of these products among both study groups. Beside salt, the most popular seasonings were mustard, lemon juice and herbs de Provence among omnivores and lemon juice, soy sauce and mustard among vegans. We found significant differences in the intake of horseradish (*p* = 0.019), herbs de Provence (*p* = 0.021), Maggi (*p* = 0.014), mustard (*p* = 0.044), lemon juice (*p* = 0.035), soy sauce (*p* = 0.000) and Vegeta (*p* = 0.008) between the study groups. Interestingly, 75% of omnivores did not consume soy sauce, citric acid, white or wine vinegar, while similar percentage of vegans did not use horseradish, Maggi, Vegeta, citric acid or white vinegar.

### 3.4. Intake of Dietary Salicylates among Omnivores and Vegans

Estimated medial salicylates intake from herbs and spices was significantly higher among vegans (*p* = 0.000) and reached 5.82 and 3.13 mg/week (831.5 vs. 447.1 µg/day) for vegans and omnivores, respectively. The main sources of salicylates were turmeric, curry, fresh garlic, rosemary and ginger among omnivores. In turn in the vegan group the main sources of salicylates were turmeric, curry, ginger, fresh garlic and rosemary (Appendix A). As shown in the Table 8, the intake of salicylates differed significantly in both study groups for 16 out of 33 condiments included in the analysis. Results from the univariate analysis have revealed that a healthy nutrition had a significant influence on the salicylates intake. Individuals with better (self-reported) nutrition consumed significantly more salicylates from herbs and spices (*p* = 0.000).

## 4. Discussion

In this study, we examined the intake of herbs and spices, as a source of dietary salicylates, among adults who follow either omnivorous or plant-based diets. As is the case with fruits and vegetables, herbs and spices are considered to be significant sources of these bioactive compounds [9,29,30,35]. To the best of our knowledge, our research is the first study to assess the intake of dietary salicylates from herbs and spices given such a large assortment of condiments. The measurement of food intake usually consumed in such small amounts is very challenging. We used FFQ to assess the intake of these products since this method was used in several studies [1,42,43,44,45,46,47] and was found to be a good tool for the assessment of consumption of specific food groups, including herbs and spices [44,47].

This study showed that one third of all 61 herbs, spices and other condiments included in the analysis was being consumed by at least 50% of participants from both groups. An average intake of condiments included in the study (except for salt, soy sauce, horseradish, lemon juice, citric acid, Maggi, white vinegar, balsamic vinegar and wine vinegar) was 15.32 ± 12.7 g/day and 18.12 ± 10.2 g/day for omnivores and vegans, respectively (*p* = 0.006). This seems to be a great value, compared to other studies, but it is worth highlighting that our analysis covered a large number of spices. Median daily intake of herbs and spices assessed by using FFQ in the study of Carlsen et al. [44] was 2.7 g/person (range 0.19 to 45.0); however, the latter study analyzed the consumption of only of 27 herbs and spices. Similarly, Siruguri and Bhat investigated the consumption of 17 spices and found that only 11 of them were frequently used [1]. Vázquez-Fresno at al. [2] reported, that the highest consumption of herbs and spices is observed in India, South Africa and Latin America (an average of 4.4 g/day), moderate intake is found in the Middle Eastern countries (2.6 g/day) and Eastern Asian countries (3.1 g/day), while the lowest consumption is observed in European countries (0.5 g/day).

The five most frequently eaten herbs in our study were parsley, garlic, dill, marjoram and basil for omnivores, while for vegans they were garlic, parsley, ginger, basil and dill. These herbs (except for ginger) are typical of the Polish cuisine. According to a CBI (Centre for the Promotion of Imports) market survey (in contrast to our results), the most popular herbs for EU market were thyme and oregano [4]. However, Szűcs et al. [47] reported that the most frequently consumed herbs and spices in seven European countries were parsley, basil, pepper and paprika. They found country specific preferences which can be still identified in Europe. These different preferences may be related to the various methods of quantifying the spices intake. CBI data are based on the consumption, which is the sum of production and importations minus exportations, and such data may have many limitations, including negative consumption (when exportations are higher than production plus importations) or high fluctuation between years. Paradoxically, European countries such as France, Italy and Greece, are important producers of dried herbs, but at the same time the consumption of these products in these countries is rather low [4].

The intake of spices varies considerably around the world. The most popular and most widely used spice in the world, called the “king of spices” is black pepper [48]. Our study confirms high frequency of intake of black pepper among omnivores and vegans; however, the median intake reached only 2 g/week/person. The leading spices consumed according to CBI data are pepper, paprika and allspice [4].

One of the interesting finding of our study is that only one (saffron) of all condiments listed on the FFQ was declared as consumed rarely or never. We can, therefore, conclude that Poles know and use herbs and spices in their cuisine. The result of Blanton [10] revealed that dill was not used by any American participant, therefore this herb could be removed from the questionnaire. Whereas, in our study dill was one of the most popular herbs among both omnivores and vegans. Our findings confirmed that the selection of herbs and spices for the study should be country-specific and consumer-specific.

The number of vegetarians is estimated to be 1% of the adult population of which 10% are vegans. In Eastern Europe, the number of vegetarians is estimated at less than 1% of the population. The vegetarian community in Poland is rather small but still growing. In 2000, around 1% of Poles considered themselves to be vegetarians. However, in 2019 this number had reached 3.7% (of which 10% were individuals aged between 23 and 34 years old) [49]. In India, approximately one-third of the population is vegetarian, while in Great Britain and Germany the proportion of vegetarians reaches 9% of the population [50,51]. Veganism is one of the fastest growing culinary trends across Europe. It is noticed that the number of vegans is increasing more rapidly than the number of vegetarians [51]. The vegan population in the Netherlands is estimated at around two percent of the population [52]. In Italy, 7.3% of the population follows a vegetarian diet of which 1.9% are vegans. The percentage of Italian vegans has tripled in the last 5 years [53].

A plant-based diet is recommended by many authorities as nutritionally adequate, appropriate for all groups of people in the prevention and management of several diseases [54,55]. It has been well recognized that the occurrence of some chronic diseases among the adult population following plant-based diet is lower [11,56,57,58]. However, some authors suggest that the existing data do not allow us to clearly evaluate the health benefits and risks of vegetarian type diets on the nutritional or health status of children and adolescents in developed countries [59,60]. These beneficial health effects of a plant-based diet may be related to different bioactive compounds, including dietary salicylates, from fruits, vegetables as well as herbs and spices. It is, nonetheless, debated whether such a diet provides a sufficient dietary intake of salicylates for it to have a therapeutic effect. Some studies suggest that vegetarians had higher circulating salicylic acid than the non-vegetarian general population, comparable to the level achieved by patients taking low dose aspirin [61,62]. Aspirin is one of the most widely used medications in the world to treat pain, fever and inflammation. It is also commonly recommended in the prevention of heart diseases and colorectal cancer [63,64,65]. Recent study supports “metabolite hypothesis”, which assumes that both aspirin and flavonoids (secondary metabolites) as well as diet may protect against colorectal and other types of cancer [66].

One of the first studies on the total salicylates content in a mixt diet suggested that it may provide 10–200 mg of salicylates/day [29]. However, subsequent studies indicated that this may be overestimated, owing to methodological differences. These studies indicated that a median salicylates intake from a mixt diet were in the range of 2 to over 4 mg/day [30,31,33]. Data from the current study have confirmed that the vegan diet contains a significantly higher level of salicylates from herbs and spices in comparison to the omnivorous diet (831.5 vs. 447.1 µg/day). Herbs and spices are highly used by people following the plant-based diet to give protein products (based on soy, chickpeas and peas protein) a more defined taste.

The typical omnivorous diet comprises food from six groups: meat and meat products, milk and dairy product, cereals, fats, fruits and vegetables. Some of these food products (i.e., highly processed food, red meat) may contain carcinogenic chemicals (i.e., heterocyclic amines, acrylamide). Herbs and spices, due to their high antioxidant activity, may suppress harmful effects of carcinogens. The highest amounts of flavonoids have been found in parsley, oregano, celery, saffron, dill and fennel [23]. Adding herbs and spices to meat dishes can also change their nutritional value. Liu et al. [67] found a significantly higher content of the free amino acids in seasoned duck compared with the unseasoned duck. Therefore, it is important to encourage people to use different kinds of fresh herbs and spices regularly, in order to give/keep flavor and to change/improve the nutritional value of dishes [13]. Replacing salt with herbs and spices can significantly reduce sodium intake and decrease the risks of heart disease and stroke among the general population. Clinical studies demonstrate that excessive sodium intake increases blood pressure [68]. Cook et al. [69] have shown that dietary sodium reduction has long-term benefits for cardiovascular health. Herb and spice blends were successfully implemented to enhance consumer acceptability of a low salt tomato soup [70]. Therefore, it seems that using herbs and spices could be an alternative method and a useful tool for reducing salt content in food products. Besides the reduction of salt intake, adding herbs and spices could be a potential means to increase the vegetable intake among adolescents and adults. However, the effectiveness of such a strategy among children and adolescents is not so evident [71,72].

### Limitations of the Study

There are some limitations to this study, which must be considered. Firstly, assessment of nutrition using self-reported data, especially concerning foods eaten in small amount, is particularly problematic and has been associated with measurement errors. Therefore, it requires replication in larger samples. Secondly, the sample was not representative and limited to the Polish population, therefore these results may not be generalized. Thirdly, we did not include the intake of herbs and spices from dishes eaten in restaurants and take-away meals, which may be important sources of these seasonings. However, according to some studies, this addition of herbs and spices to dishes represents only 0.5%–1.0% [44]. Finally, it is worth highlighting that the data on the content of salicylates in herbs and spices are limited and may also constitute limitations of this study.

## 5. Conclusions

In conclusion, our study confirmed that herbs and spices are important sources of salicylates; however, the type of diet influences their level in the diet. Our study revealed that the median of daily salicylates intake was significantly higher among vegans than omnivores. These observations allow us to conclude that herbs and spices, although consumed in small quantities, may still be important contributors of salicylates. In combination with other sources of salicylates (fruits, vegetables, some cereals and beverages) and antioxidant-rich foods, they can give the plant-based diet therapeutic properties. This could be an attractive and safe approach to disease prevention and management, requiring further research. Furthermore, dietary guidelines for the general population, which emphasize herbs and spices consumption, should be recommended in addition to calories control, portion size control and increased physical activity.

## Figures and Tables

**Figure 1 nutrients-12-02727-f001:**
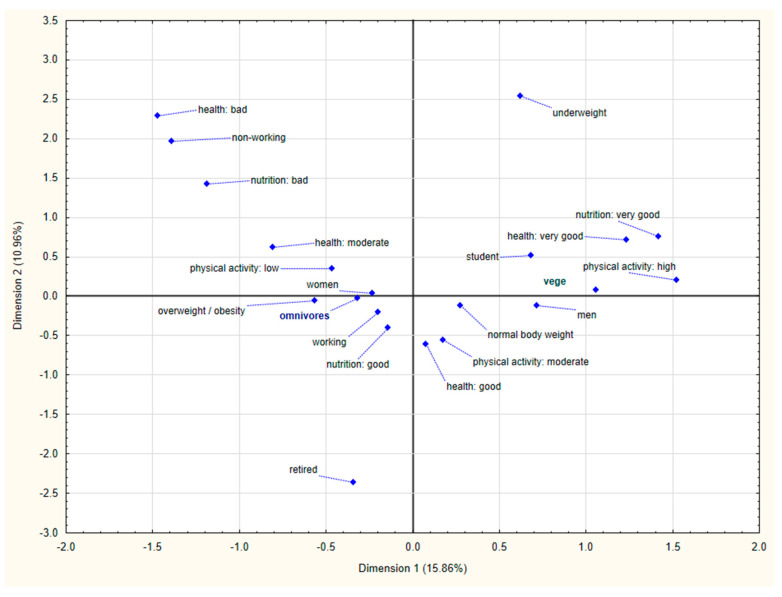
Characteristics of adult omnivores and vegans according to the parameters concerning gender, body mass, health status, physical activity and self-reported diet quality (the correspondence analysis).

**Table 1 nutrients-12-02727-t001:** Characteristics of the study population.

Parameter	Omnivores n = 208	Vegans n = 62	*p* Value *
Women, n (%)	171 (82.2)	34 (54.8)	
Age (years)	37.0 ± 12.1 ^1^	28.11 ± 7.7	0.000 ^2^
BMI (kg/m^2^)	24.6 ± 4.1	22.7 ± 3.3	0.001 ^2^
Body weight status, n (%)			
Underweight	6 (2.9)	4 (6.5)	0.021 ^3^
Normal body weight	121 (58.2)	45 (72.6)	
Overweight	56 (26.9)	13 (19.3)	
Obesity	25 (12.0)	1 (1.6)	
Employment status, n (%)			
Student	43 (20.7)	24 (38.7)	0.033 ^3^
Working	159 (76.5)	37 (59.7)	
Non-working	4 (1.9)	1 (1.6)	
Retired	2 (0.9)	0	
Physical activity, n (%)			
Low	118 (56.7)	22 (35.5)	0.001 ^3^
Moderate	73 (35.1)	26 (41.9)	
High	17 (8.2)	14 (22.6)	
Health status, n (%)			
Bad	3 (1.4)	0	0.029 ^3^
Moderate	66 (31.7)	12 (19.4)	
Good	111 (53.4)	33 (53.2)	
Very good	28 (13.5)	17 (27.4)	
Nutrition, n (%)			
Bad	31 (14.9)	0	0.000 ^3^
Good	152 (73.1)	42 (67.7)	
Very good	25 (12.0)	20 (32.3)	

^1^ mean ± standard deviation; ^2^ the Mann–Whitney U test; ^3^ the chi-square Pearson test; * *p*-value—differences between omnivores and vegans.

**Table 2 nutrients-12-02727-t002:** Frequency of consumption of fresh herbs among adult omnivores and vegans.

Fresh Herbs	Frequency of Consumption (%) ^1^												
	Daily		Weekly						Monthly		Rarely or Never		*p* Value ^2^
	1–3		5–6		3–4		1–2		1–3				
	O	V	O	V	O	V	O	V	O	V	O	V	
Basil	7.2	12.9	2.4	4.8	12.5	17.7	21.6	22.6	20.7	12.9	35.6	29.0	0.318
Chili	3.9	0.0	2.9	0.0	13.0	4.8	14.9	6.5	8.2	11.3	57.2	77.4	0.020
Coriander	1.0	3.2	1.0	4.8	6.7	6.5	1.9	14.5	3.4	16.1	86.1	54.8	0.000
Dill	12.5	3.2	6.7	3.2	17.8	12.9	24.0	30.7	11.1	16.1	27.9	33.9	0.154
Garlic	9.6	21.0	7.2	14.5	26.0	29.0	28.9	24.2	12.0	11.3	16.4	0.0	0.002
Ginger	4.8	11.3	1.4	3.2	5.8	27.4	11.1	21.0	12.0	9.7	64.9	27.4	0.000
Lovage	2.9	0.0	2.4	0.0	5.3	1.6	6.3	2.6	7.2	3.2	76.0	93.6	0.082
Marjoram	9.6	1.6	3.9	0.0	13.5	0.0	21.6	8.1	13.9	21.0	37.5	69.4	0.000
Melissa	4.3	0.0	1.0	0.0	5.3	3.2	8.7	4.8	9.1	21.0	71.6	71.0	0.069
Mint	7.7	3.2	2.9	1.6	7.7	8.1	15.4	11.3	13.9	21.0	52.4	54.8	0.570
Oregano	7.2	4.8	2.4	3.2	9.6	8.1	18.3	14.5	14.4	17.7	48.1	51.6	0.912
Parsley	15.9	25.8	5.8	9.7	19.7	19.4	25.0	17.7	15.4	11.3	18.3	16.1	0.363
Rosemary	2.9	1.6	1.0	1.6	6.7	4.8	7.7	9.7	7.7	12.9	74.0	69.4	0.764
Sage	1.4	0.0	1.4	0.0	1.9	0.0	2.9	0.0	3.4	9.7	88.9	90.3	0.120
Tarragon	1.9	0.0	0.0	0.0	4.8	0.0	2.9	0.0	4.3	3.2	86.1	96.8	0.151
Thyme	3.4	9.9	1.4	1.6	6.3	6.5	9.1	8.1	9.6	17.7	70.2	66.1	0.413

O—omnivores; V—vegans; ^1^ percentage of respondents; ^2^
*p*-value of the chi-square Pearson test between groups.

**Table 3 nutrients-12-02727-t003:** Weekly intake of fresh herbs among adult omnivores and vegans.

Fresh Herbs	Weekly Intake of Fresh Herbs (g)												
	Omnivores						Vegans						
	Mean	SD	Q1	Q2	Q3	Q4	Mean	SD	Q1	Q2	Q3	Q4	*p* Value ^1^
Basil	1.39	2.93	0.00	0.39	1.25	23.47	6.99	8.83	0.50	3.12	11.43	29.09	0.000
Chili	0.90	1.99	0.00	0.00	0.80	10.69	0.51	1.29	0.0	0.0	0.18	5.34	0.156
Coriander	0.35	1.24	0.00	0.00	0.00	5.82	0.54	0.95	0.00	0.10	0.55	4.38	0.000
Dill	1.63	2.82	0.00	0.81	2.25	17.41	0.92	1.89	0.04	0.36	0.97	13.52	0.105
Garlic	34.64	37.29	7.30	23.48	54.79	178.46	49.75	36.07	23.48	49.31	86.09	109.57	0.001
Ginger	7.59	17.18	0.00	0.00	6.90	105.22	27.16	28.09	1.24	22.19	51.77	103.53	0.000
Lovage	0.52	1.60	0.00	0.00	0.00	10.59	0.08	0.43	0.00	0.00	0.00	3.32	0.416
Marjoram	1.12	2.50	0.00	0.24	0.97	13.60	0.26	1.20	0.00	0.00	0.21	9.47	0.000
Melissa	0.29	0.80	0.00	0.00	0.12	6.97	0.10	0.23	0.00	0.00	0.12	1.16	0.634
Mint	0.75	1.65	0.00	0.00	0.71	11.67	0.49	1.23	0.00	0.04	0.28	8.43	0.851
Oregano	1.05	2.03	0.00	0.20	1.33	14.44	0.83	1.92	0.00	0.08	0.67	12.42	0.924
Parsley	1.78	2.19	0.18	1.13	2.64	10.56	2.47	2.77	0.35	1.71	4.15	15.84	0.051
Rosemary	2.07	6.52	0.00	0.00	0.62	42.59	1.57	3.91	0.00	0.00	1.24	21.30	0.066
Sage	0.08	0.35	0.00	0.00	0.00	3.04	0.02	0.05	0.0	0.00	0.00	0.20	0.368
Tarragon	0.13	0.49	0.00	0.00	0.00	3.09	0.01	0.05	0.00	0.00	0.00	0.29	0.804
Thyme	0.73	1.65	0.00	0.00	0.52	8.27	0.53	1.09	0.00	0.00	0.52	4.13	0.231

SD—standard deviation; Q1—25th percentile; Q2—median (50th percentile); Q3—75th percentile; Q4—maximum; ^1^
*p*-value of the Mann–Whitney U test between groups.

**Table 4 nutrients-12-02727-t004:** Frequency of consumption of dried spices among adult omnivores and vegans.

Dried Spices	Frequency of Consumption (%) ^1^												
	Daily		Weekly						Monthly		Rarely or Never		*p* Value ^2^
	1–3		5–6		3–4		1–2		1–3				
	O	V	O	V	O	V	O	V	O	V	O	V	
Anise	1.0	1.6	0.5	0.0	0.0	0.0	0.5	0.0	2.9	4.8	95.2	93.6	0.855
Allspice	16.8	1.6	8.2	1.6	13.0	11.3	23.1	14.5	12.5	14.5	26.4	56.5	0.000
Basil	8.7	6.5	2.9	8.1	13.0	19.4	17.8	17.7	13.5	17.7	44.2	30.7	0.185
Bay leaf	11.1	3.2	6.7	4.8	19.7	9.7	24.5	14.5	15.4	37.1	22.6	30.7	0.001
Black cumin	1.4	0.0	1.9	0.0	2.4	0.0	4.8	4.8	7.2	12.9	82.2	82.3	0.370
Black pepper	38.0	30.7	18.8	21.0	15.4	17.7	9.1	14.5	2.9	1.6	15.9	14.5	0.749
Caraway	1.4	1.6	1.4	0.0	3.9	4.8	3.9	1.6	6.3	17.7	83.2	74.2	0.098
Cardamom	1.0	1.6	0.5	0.0	2.9	0.0	2.4	4.8	7.7	12.9	85.6	80.7	0.438
Cayenne pepper	6.3	11.3	3.9	6.5	10.6	11.3	5.8	14.5	4.8	17.7	68.8	38.7	0.000
Celery	5.8	1.6	1.9	0.0	4.3	6.5	9.6	8.1	6.7	1.6	71.6	82.3	0.264
Cinnamon	5.8	4.8	4.3	0.0	5.8	4.8	14.4	21.0	12.5	12.9	57.2	56.5	0.531
Cloves	2.4	3.2	1.9	6.5	2.4	9.7	5.8	8.1	11.5	24.2	76.0	48.4	0.001
Cocoa	8.2	4.8	1.9	8.1	8.7	11.3	21.6	21.0	14.9	21.0	44.7	33.9	0.113
Cumin	0.5	0.0	0.5	0.0	1.0	14.5	3.9	12.9	4.8	9.7	89.4	62.9	0.000
Fenugreek	0.5	1.6	0.0	0.0	1.4	0.0	1.9	0.0	1.4	4.8	94.7	93.6	0.249
Flax seeds	2.4	11.3	1.9	4.8	2.4	12.9	9.6	17.7	4.3	12.9	79.3	40.3	0.000
Garlic	5.3	0.0	6.7	8.7	2.9	9.7	11.5	9.7	11.1	4.8	62.5	67.7	0.059
Ginger	2.9	0.0	2.9	0.0	5.3	3.2	10.1	4.8	13.5	8.1	65.4	83.9	0.110
Hot paprika	7.2	1.6	1.9	6.5	17.3	11.3	19.2	12.9	9.6	4.8	44.7	62.9	0.028
Juniper	1.4	1.6	0.5	0.0	1.0	0.0	0.5	0.0	2.9	0.0	93.8	98.4	0.685
Lovage	2.4	0.0	1.4	0.0	7.2	6.5	8.2	6.5	7.2	4.8	73.6	82.3	0.625
Mustard	1.9	0.0	1.4	0.0	1.4	0.0	0.5	1.6	10.6	6.5	84.1	91.9	0.417
Nutmeg	3.4	0.0	2.4	0.0	1.4	3.2	4.3	6.5	18.3	16.1	70.2	74.2	0.407
Onion	1.9	0.0	1.4	3.2	3.9	0.0	5.8	11.3	4.3	9.7	83.7	75.8	0.122
Oregano	5.8	3.2	3.9	4.8	14.4	24.2	21.2	24.2	11.5	14.5	43.3	29.0	0.268
Rosemary	1.9	0.0	2.9	4.8	5.8	12.9	8.7	12.9	9.6	12.9	71.2	56.5	0.160
Saffron	0.0	0.0	0.5	0.0	1.0	0.0	1.0	0.0	2.0	0.0	95.5	100	0.735
Savory	1.4	0.0	0.5	0.0	1.9	1.6	1.4	4.8	2.9	8.1	91.8	85.5	0.216
Sweet paprika	9.6	3.2	7.7	11.3	22.1	29.0	21.6	12.9	12.5	11.3	26.4	32.3	0.254
Turmeric	2.4	9.7	2.4	0.0	7.2	16.1	12.5	32.3	13.0	16.1	62.5	25.8	0.000
White pepper	7.2	0.0	2.4	0.0	5.8	0.0	8.7	8.1	3.4	9.7	72.6	82.3	0.015

O—omnivores; V—vegans; ^1^ percentage of respondents; ^2^
*p*-value of the chi-square Pearson test between groups.

**Table 5 nutrients-12-02727-t005:** Weekly intake of dried spices among adult omnivores and vegans.

Dried Spices	Weekly Intake of Dried Spices (g)												
	Omnivores						Vegans						
	Mean	SD	Q1	Q2	Q3	Q4	Mean	SD	Q1	Q2	Q3	Q4	*p* Value ^1^
Anise	0.15	1.37	0.00	0.00	0.00.	19.05	0.39	2.29	0.00	0.00	0.00	17.96	0.212
Allspice	4.44	7.20	0.00	1.73	4.79	36.61	0.97	1.60	0.00	0.14	1.26	6.10	0.000
Basil	1.24	2.55	0.00	0.29	1.08	14.78	1.33	1.96	0.85	0.66	2.41	12.25	0.004
Bay leaf	5.69	9.42	0.51	2.71	6.32	53.08	1.11	1.61	0.09	0.34	1.19	5.96	0.000
Black cumin	0.40	1.49	0.00	0.00	0.00	11.85	0.14	0.31	0.00	0.00	0.10	1.51	0.053
Black pepper	3.39	4.71	0.96	1.89	4.24	32.39	3.07	4.62	0.58	2.14	2.72	32.39	0.911
Caraway	0.44	1.56	0.00	0.00	0.00	11.14	0.40	1.14	0.00	0.00	0.31	7.25	0.008
Cardamom	0.51	1.86	0.00	0.00	0.00	11.82	0.35	0.75	0.00	0.00	0.30	3.79	0.006
Cayenne pepper	0.78	1.89	0.00	0.00	0.45	11.31	1.16	2.16	0.00	0.24	1.60	11.31	0.000
Celery	1.24	3.59	0.00	0.00	0.19	23.09	0.53	1.73	0.00	0.00	0.00	11.55	0.306
Cinnamon	0.93	2.00	0.00	0.00	0.71	14.54	1.19	2.06	0.00	0.00	0.99	7.96	0.312
Cloves	0.69	2.75	0.00	0.00	0.00	23.39	1.21	1.98	0.00	0.43	1.67	7.80	0.000
Cocoa	4.15	10.41	0.00	0.74	3.70	70.73	2.22	2.97	0.10	0.75	2.39	11.14	0.145
Cumin	0,16	0.80	0.00	0.00	0.00	7.78	0.80	1.64	0.00	0.05	0.68	7.78	0.000
Fenugreek	0.13	0.70	0.00	0.00	0.00	5.20	0.05	0.21	0.00	0.00	0.00	1.31	0.858
Flax seeds	0.81	2.57	0.00	0.00	0.00	20.56	4.69	6.85	0.00	1.38	5.05	20.56	0.000
Garlic	1.41	4.07	0.00	0.00	0.99	41.92	1.01	2.07	0.00	0.00	1.08	9.02	0.844
Ginger	0.47	1.29	0.00	0.00	0.28	9.73	0.18	0.62	0.00	0.00	0.00	4.26	0.143
Hot paprika	1.42	2.45	0.00	0.29	2.14	12.84	0.97	2.13	0.00	0.00	0.92	10.39	0.097
Juniper	0.15	1.04	0.00	0.00	0.00	12.79	0.15	1.08	0.00	0.00	0.00	8.54	0.913
Lovage	0.40	1.39	0.00	0.00	0.11	10.37	0.15	0.46	0.00	0.00	0.03	2.96	0.722
Mustard	0.39	1.85	0.00	0.00	0.00	18.91	0.11	0.54	0.00	0.00	0.00	4.05	0.969
Nutmeg	0.43	1.37	0.00	0.00	0.20	10.74	0.21	0.54	0.00	0.00	0.20	2.73	0.712
Onion	0.53	2.01	0.00	0.00	0.00	14.57	0.59	1.77	0.00	0.00	0.13	12.21	0.164
Oregano	0.62	0.96	0.00	0.19	0.81	6.07	0.70	0.81	0.03	0.53	0.82	3.79	0.020
Rosemary	0.56	1.39	0.00	0.00	0.28	8.53	0.85	1.39	0.00	0.10	0.91	4.57	0.000
Saffron	0.01	0.09	0.00	0.00	0.00	0.88	0.00	0.02	0.00	0.00	0.00	0.19	0.830
Savory	0.11	0.57	0.00	0.00	0.00	5.83	0.10	0.28	0.00	0.00	0.00	1.46	0.056
Sweet paprika	2.17	2.84	0.00	0.88	2.93	12.01	2.19	3.00	0.07	0.86	3.15	13.09	0.839
Turmeric	0.79	1.76	0.00	0.00	0.94	11.06	1.76	3.09	0.24	1.04	2.42	14.08	0.000
White Pepper	1.03	2.59	0.00	0.00	0.30	13.43	0.21	0.60	0.00	0.00	0.11	4.02	0.701

SD—standard deviation; Q1—25th percentile; Q2—median (50th percentile); Q3—75th percentile; Q4—maximum; ^1^
*p*-value of the Mann–Whitney U test, between groups.

**Table 6 nutrients-12-02727-t006:** Frequency of consumption of seasoning blends and other spices among adult omnivores and vegans.

Condiment	Frequency of Consumption (%) ^1^												
	Daily		Weekly						Monthly		Rarely or Never		*p* Value ^2^
	1–3		5–6		3–4		1–2		1–3				
	O	V	O	V	O	V	O	V	O	V	O	V	
Curry spice blend	6.3	3.2	3.9	1.6	7.7	8.1	11.5	19.4	11.5	8.1	59.1	59.7	0.516
Horseradish	5.3	0.0	2.4	1.6	6.7	1.6	13.9	3.2	11.1	1.6	60.6	91.9	0.000
Herbs de Provence	11.5	6.5	4.8	1.6	13.0	4.8	19.7	9.7	12.5	14.5	38.5	62.9	0.012
Maggi	8.2	0.0	3.4	0.0	3.9	0.0	6.7	6.5	3.9	0.0	74.0	93.6	0.014
Mustard	12.0	3.2	5.8	3.2	18.8	12.9	27.9	24.2	13.9	17.7	21.6	38.7	0.044
Ready-mix spices to dishes	8.2	0.0	1.9	3.2	11.5	4.8	6.7	8.1	9.6	14.5	62.0	69.4	0.103
Soy sauce	3.4	9.7	1.0	6.5	3.9	12.9	4.8	22.6	10.6	17.7	76.4	30.7	0.000
Vegeta	6.7	1.6	2.9	0.0	6.3	0.0	9.1	1.6	6.7	3.2	68.3	93.6	0.005
Salt	60.1	62.9	14.9	12.9	8.2	12.9	8.7	3.2	1.9	0.0	6.3	8.1	0.457
Lemon juice	24.0	19.4	8.2	9.7	18.3	9.7	13.0	30.7	10.6	9.7	26.0	21.0	0.035
Citric acid	1.0	0.0	1.4	0.0	2.9	1.6	7.2	3.2	9.1	0.0	78.4	95.2	0.067
White vinegar	2.9	0.0	0.5	0.0	1.4	0.0	4.3	3.2	11.1	1.6	79.8	95.2	0.099
Balsamic vinegar	1.0	0.0	1.9	0.0	3.9	3.2	7.7	9.7	11.5	6.5	74.0	80.7	0.618
Wine vinegar	1.9	0.0	3.9	0.0	1.4	6.5	5.8	8.1	7.7	12.9	79.3	72.6	0.066

O—omnivores; V—vegans; ^1^ percentage of respondents; ^2^
*p*-value of the chi-square Pearson test between groups.

**Table 7 nutrients-12-02727-t007:** Weekly intake of seasoning blends and other spices among adult omnivores and vegans.

Condiment	Weekly Intake of Seasoning Blends and Other Spices (g)												
	Omnivores						Vegans						
	Mean	SD	Q1	Q2	Q3	Q4	Mean	SD	Q1	Q2	Q3	Q4	*p* Value ^1^
Curry spice blend	2.31	5.92	0.00	0.00	2.00	44.32	1.74	3.25	0.00	0.00	2.00	14.77	0.513
Horseradish	2.97	7.18	0.00	0.00	2.18	49.15	0.94	3.93	0.00	0.00	0.00	25.75	0.019
Herbs de Provence	1.95	3.57	0.00	0.53	2.73	22.29	1.24	3.24	0.00	0.09	0.53	15.89	0.021
Maggi	3.62	12.08	0.00	0.00	0.25	75.31	0.35	1.33	0.00	0.00	0.00	5.38	0.042
Mustard	15.09	20.22	0.21	6.71	15.65	104.13	11.51	15.56	0.00	4.85	15.49	72.28	0.504
Ready-mix spices to dishes	3.89	8.70	0.00	0.00	2.85	52.64	1.55	3.78	0.00	0.00	1.77	20.68	0.462
Soy sauce	1.14	3.49	0.00	0.00	0.00	26.92	5.63	6.98	0.18	2.27	10.57	21.14	0.000
Vegeta	4.68	14.81	0.00	0.00	2.38	112.70	0.62	3.95	0.00	0.00	0.00	30.95	0.008
Salt	16.42	17.14	4.35	9.73	22.10	83.10	24.21	18.93	11.05	19.80	35.00	66.30	0.001
Lemon juice	11.37	17.36	0.00	4.87	17.50	73.40	9.73	11.29	1.64	5.24	19.22	48.93	0.235
Citric acid	0.40	1.52	0.00	0.00	0.00	16.21	0.10	0.47	0.00	0.00	0.00	3.21	0.161
White vinegar	0.49	1.87	0.00	0.00	0.00	14.09	0.10	0.54	0.00	0.00	0.00	3.02	0.162
Balsamic vinegar	1.16	3.65	0.00	0.00	0.16	24.68	0.77	2.07	0.00	0.00	0.21	12.34	0.638
Wine vinegar	1.03	3.32	0.00	0.00	0.00	18.91	0.94	2.40	0.00	0.00	0.41	9.46	0.128

SD—standard deviation; Q1—25th percentile; Q2—median (50th percentile); Q3—75th percentile; Q4—maximum; ^1^
*p*-value of the Mann–Whitney U test between groups.

**Table 8 nutrients-12-02727-t008:** Dietary salicylates intake from herbs, spices and other condiments among adult omnivores and vegans.

Herbs, Spices or Other Condiments		Salicylates Intake (µg/week)												
		Omnivores						Vegans						
		Mean	SD	Q1	Q2	Q3	Q4	Mean	SD	Q1	Q2	Q3	Q4	*p* Value ^1^
Fresh herbs	Basil	4.49	9.48	0.00	1.26	4.04	76.06	22.63	28.59	1.62	10.10	37.03	94.26	0.000
	Chili	5.88	13.09	0.00	0.00	5.27	70.23	3.36	8.49	0.00	0.00	1.20	35.12	0.156
	Dill	5.24	9.08	0.00	2.60	7.25	56.01	2.97	6.07	0.12	1.15	3.11	43.49	0.105
	Garlic	590.54	635.72	124.55	400.33	934.10	3042.71	848.31	615.04	400.33	840.69	1467.87	1868.20	0.001
	Ginger	239.72	542.45	0.00	0.00	217.90	3321.94	857.44	886.64	39.14	700.38	1634.21	3268.42	0.000
	Mint	70.89	155.05	0.00	0.00	66.46	1097.15	45.75	115.34	0.00	3.40	26.61	792.63	0.851
	Parsley	4.97	6.13	0.50	3.17	7.39	29.57	6.92	7.75	0.993	4.79	11.62	44.35	0.051
Dried spices	Allspice	221.81	359.95	0.00	86.69	239.72	1830.47	48.43	79.85	0.00	6.94	63.08	305.08	0.000
	Basil	136.52	279.91	0.00	31.58	118.42	1623.99	146.40	215.78	9.28	72.07	264.26	1345.32	0.004
	Bay leaf	143.26	237.27	12.74	68.25	159.25	1337.66	27.92	40.58	2.18	8.55	29.94	150.15	0.000
	Black pepper	8.44	11.74	2.35	4.70	10.57	80.68	7.65	11.52	1.45	5.32	6.78	80.68	0.911
	Caraway	0.13	0.44	0.00	0.00	0.00	3.17	0.11	0.32	0.00	0.00	0.09	2.07	0.008
	Cardamom	67.27	244.81	0.00	0,00	0,00	1560.17	46.21	99.39	0.00	0.00	40.03	500.41	0.006
	Cinnamon	8.33	18.00	0.00	0.00	6.42	130.94	10.72	18.57	0.00	0.00	8.89	71.70	0.312
	Cloves	15.07	60.49	0.00	0.00	0.00	514.17	26.49	43.50	0.00	9.55	36.73	171.39	0.000
	Cocoa	2.29	5.74	0.00	0.41	2.04	38.96	1.22	1.64	0.05	0.41	1.31	6.14	0.145
	Cumin	11.05	54.07	0.00	0.00	0.00	524.73	54.23	110.92	0.00	3.66	45.81	524.73	0.000
	Fenugreek	0.13	0.70	0.00	0.00	0.00	5.20	0.05	0.21	0.00	0.00	0.00	1.31	0.858
	Garlic	0.48	1.39	0.00	0.00	0.34	14.34	0.34	0.71	0.00	0.00	0.37	3.09	0.844
	Ginger	0.55	1.51	0.00	0.00	0.33	11.38	0.20	0.72	0.00	0.00	0.00	4.98	0.143
	Hot pepper	8.44	14.59	0.00	1.70	12.74	76.46	5.77	12.69	0.00	0.00	5.46	61.87	0.097
	Nutmeg	10.35	32.92	0.00	0.00	4.68	257.84	5.11	12.84	0.00	0.00	4.72	65.48	0.712
	Oregano	28.41	43.69	0.00	8.75	37.04	277.14	31.88	37.01	1.28	24.09	37.48	172.88	0.020
	Rosemary	380.64	947.24	0.00	0.00	193.45	5802.92	578.32	943.63	0.00	67.87	621.74	3110.93	0.000
	Sweet pepper	7.84	10.26	0.00	3.19	10.60	43.40	7.90	10.85	0.25	3.10	11.37	47.32	0.839
	Turmeric	2765.26	6175.17	0.00	0.00	3289.00	38,763.20	6176.59	10,842.21	845.74	3630.83	8471.94	49334.98	0.000
	White pepper	0.48	1.20	0.00	0.00	0.14	6.21	0.10	0.28	0.00	0.00	0.05	1.86	0.701
Other condiments	Balsamic vinegar	0.16	0.50	0.00	0.00	0.02	3.37	0.10	0.28	0.00	0.00	0.03	1.69	0.638
	Curry	1719.41	4410.32	0.00	0.00	1491.73	33,038.74	1298.17	2421.49	0.00	0.00	1491.73	11,012.91	0.513
	Lemon	0.63	0.96	0.00	0.27	0.97	4.07	0.54	0.63	0.09	0.29	1.07	2.72	0.235
	Maggi	0.22	0.74	0.00	0.00	0.02	4.64	0.02	0.08	0.00	0.00	0.00	0.33	0.042
	Soy sauce	0.66	2.02	0.00	0.00	0.00	15.58	3.26	4.04	0.10	1.31	6.12	12.24	0.000
	Vegeta	5.75	18.19	0.00	0.00	2.92	138.41	0.77	4.85	0.00	0.00	0.00	38.01	0.008
Total salicylates		6444.08	9630.94	825.20	3129.87	7066.86	62,524.38	10,237.12	12411.03	3423.57	5820.70	12,212.64	55,571.52	0.000

SD—standard deviation; Q1—25th percentile; Q2—median (50th percentile); Q3—75th percentile; Q4—maximum; ^1^
*p*-value of the Mann–Whitney test between groups

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
