# Peer review of "Intake of Dietary Salicylates from Herbs and Spices among Adult Polish Omnivores and Vegans"

_nutrients, 2020, doi:10.3390/nu12092727_

Round 1
Reviewer 1 Report
The authors present data on the intake of salicylates by omnivores and vegans with regard to the effects of these bioactive compounds (the term “therapeutic properties, i.e. line 389/390 is not appropriate with regard to preventive effect of foods).
This idea is not reflected by the introduction which is too general and comprises a lot of information concerning the intake of different herbs and spices all over the world. The term “salicylates” is mentioned for the first time in line 57 (“including salicylates”) but no information is given on the specific significance of these compounds. Thus, the reader does not get an idea why the author focus on salicylates and why these substances are meaningful for human health.
Beyond this background the introduction should be shortened and rewritten. Unnecessary details should be deleted whereas the scientific question should be adequately addressed by explaining the specific meaning of salicylates.
The paper is an extensive data collection but the methods raise many questions and it is doubtful whether the results are valid. The author state that a “validated semi-quantitative food frequency questionnaire” was used (line 138) to assess the intake of herbs and spices. It is well known that data from FFQ should be regarded with caution especially in the case of food consumed in (very) small amounts. For this reason, the results presented in Table 5 and 7 (and consequently in Table 8) suggest an exactness that is not reflected by the methods applied.
Furthermore, for calculation of salicylate intake the investigators “have used several databases” (line 86/87) including data obtained by themselves. As an appropriate analysis of salicylates comprises several problems the validity of these data remains unclear. At least, an aggregation of the different data is not justified without validation.
Finally, the participants were asked to give information of a lot of herbs and spices in detail. Is it really sure that all of them knew all of them? This must be questioned.
In sum the paper presents a lot of exact calculations on doubtful data. In Table 8, for example, salicylate intakes of less then 1 mg per week are presented – it is obvious that so small amounts based on the methods applied are not sound.
What´s about standard deviations?
Minor remarks:
Line 52: „of strong flavors“, delete one time
Line 60/61: bioactive food compounds should not be generally called “nutraceuticals”, thus delete.
Line 350: Is it really a “range of 2 g to over 4 mg/day”? Typo?
Author Response
Dear Reviewers,
We are very grateful for the review provided by the external reviewers of this manuscript. Your comments provided valuable insights to refine its content. All added changes are highlighted in yellow in the text.

Reviewer 2 Report
This reviewer thanks the Nutrients Journal, for the opportunity to review this manuscript on the Intake of salicylates in Polish Omnivores and Vegans.
Veganism has been hailed as a healthy lifestyle choice and rapidly being adapted worldwide. Phytochemicals/bioactive components present in vegetables and fruits are considered to play a crucial role in myriad of health benefits. However, the role of habitually consumed herbs and spices in health contributions is not clear. In addition, it is not known about the contribution of herbs and spices to total salicylates from dietary sources. This manuscript explores the correlation between herb and spice consumption and salicylates among Polish population. Using specially designed semi quantitative Food Frequency Questionnaires author’s team collected habitual consumption of 61 fresh and dried herbs and spices during the preceding three months. They determine salicylate consumption by correlating FFQ data from Omnivores and Vegans with known database on food composition including salicylates. They identify parsley, garlic, dill, marjoram and basil as five most frequently consumed herbs by Omnivores whereas vegans consumed garlic, parsley, ginger, basil and dill. Further they demonstrate that herbs and spices serve as reliable sources for salicylates and that Vegans consume substantially higher levels of salicylates compared to omnivores.
This reviewer has a few concerns that need to be addressed, before the manuscript can be further considered.
- Vegan group appears to be relatively younger than Omnivore group. This difference may play a role in their lifestyle patterns including physical activity, dietary habits and choices. Omnivores being in the older group and may have other co-morbidities that are usually rare for the younger Vegan group. Do these differences in age groups skew the observations?
- Based on the data in the tables 2 and 3; for the fresh herb consumption, only differences between both the groups is that Vegans consume higher levels of coriander and ginger whereas Omnivores consume higher levels of marjoram. The least consumed herbs were tarragon, lovage and sage among Vegans and sage, tarragon and coriander among omnivores. There appears to be certain preferential consumption of spices by Omnivores more frequently than the Vegans. Without information on the salicylate content of major herbs based on current databases, it will be harder to interpret the results. It will be helpful if a list of salicylates containing herbs and spices (high and low salicylate containing herbs) is included in the manuscript.
- Based on differences in herb and spice consumption, it is apparent that Vegans and Omnivores have different food preferences and/or consume different foods. It will be helpful if a brief description is included about what types of dishes were (such as steaks, pasta, or noodles etc) consumed by omnivores and vegan participants.
- Depending on whether it is a pre-prepared food or freshly made dish, types of herbs and spices used in the preparation may vary. It will be helpful if information about whether processed foods or freshly prepared foods are consumed by either of these groups.
- There is no description of what ‘allspice’ includes.
- It is not entirely clear if the ultimate goal of this study is to make a case for encouraging higher consumption of herbs and spices or just foods rich in salicylates.
- Considering that even within India, type of spice consumption varies widely among Northern and Southern populations, it is not clear if the results from Southern Indian populations can be generalized to Polish populations.
- Some of the herb and spice use is cultural and region specific. Considering the diversity of spice consumption across cultures and regions, generalizability of these observations is somewhat limited to Polish population and that should be acknowledged as one of the limitations.
- Lack of objective measurement of salicylates is a limitation to the study. It should be acknowledged as well.
- In the Table 1; it is not clear what criteria were used for determining health and nutrition statuses.
- The p-value for Garlic consumption was represented as (0.001) in the text and (0.002) in the table 2. It is not clear, which one is correct.
- In the table 1 for the employment status % was listed as 59,7. It should be 7 (use a decimal, not a comma).
- Supplemental material could not be accessed from the link provided.

Author Response

(The authors gave the same response as above.)

Reviewer 3 Report
I felt the methodology was well described and the limitations of the study and the study answers the aims of it. The aims are a bit lost in the text.
The authors described these types of study are never as accurate as ones including weighed food intakes! The English is good and as already commented above verbose. The aims are clear but perhaps lost in the text. The authors could have put more about allergies and salicylates.
There is a detail of measurements eg for a pinch --but due to finger size may vary
Author Response

(The authors gave the same response as above.)
